# 3,4-Enhanced Polymerization of Isoprene Catalyzed by Side-Arm Tridentate Iminopyridine Iron Complex with High Activity: Optimization via Response Surface Methodology

**DOI:** 10.3390/polym15051231

**Published:** 2023-02-28

**Authors:** Zhenyu Han, Yongqiang Zhang, Liang Wang, Guangqian Zhu, Jia Kuang, Guangyu Zhu, Guangqiang Xu, Qinggang Wang

**Affiliations:** Key Laboratory of Biobased Materials, Qingdao Institute of Bioenergy and Bioprocess Technology, Chinese Academy of Sciences, Qingdao 266101, China

**Keywords:** tridentate iminopyridine iron complex, high activity, isoprene polymerization, response surface methodology

## Abstract

3,4-Enhanced polymerization of isoprene catalyzed by late transition metal with high activity remains one of the great challenges in synthetic rubber chemistry. Herein, a library of [N, N, X] tridentate iminopyridine iron chloride pre-catalysts (**Fe 1–4**) with the side arm were synthesized and confirmed by the element analysis and HRMS. All the iron compounds served as highly efficient pre-catalysts for 3,4-enhanced (up to 62%) isoprene polymerization when 500 equivalent MAOs were utilized as co-catalysts, delivering the corresponding high-performance polyisoprenes. Furthermore, optimization via single factor and response surface method, it was observed that the highest activity was obtained by complex **Fe 2** with 4.0889 × 10^7^ g·mol(Fe)^−1^·h^−1^ under the following conditions: Al/Fe = 683; IP/Fe = 7095; t = 0.52 min.

## 1. Introduction

Polymer with specific selectivity is one of the most essential substances in human society, and its excellent properties are highly related to the microstructure [1,2,3]. For example, natural rubber with >99% *cis*-1,4-configuration is used as an important fundamental material in tire manufacture, while gutta-percha with high *trans*-1,4-selectivity has distinctive application in insulation and medicinal materials [4,5,6]. Additionally, the artificial 3,4-polyisoprene has been demonstrated as a high-performance rubber material with prominent nature of wet skid resistance, with comprehensive properties compared with solution-polymerized butadiene styrene rubber (SSBR) [7,8,9,10]. Therefore, extensive research works have been dedicated to cultivating high selective transition metal catalytic systems via reasonable ligands, such as rare-earth [11,12], cobalt [13,14], titanium [15,16], and iron [17,18]. Among these catalytic systems, the iron complex possessed the advantages of adequate source, low price, and environmental friendliness, which attracted more and more attention [19,20,21].

Since Brookhart et al. [22] prepared N,N,N-α-diimine-FeCl_2_ complexes which showed amazing ethylene polymerization performance when activated by methylaluminoxane (MAO), more and more researchers had devoted to the study of conjugated dienes polymerization by a well-defined iron catalyst, especially in iminopyridine iron complex [23]. Up to date, well-defined iron catalysts for the low-3,4-selective (3,4-content <20%) and moderate 3,4-selective (3,4-content 20~50%) polymerization with moderate activity have been extensively reported (Figure 1). For example, Ritter et al. [24] reported a milestone work that a well-defined iminopyridine iron complex activated by alkyl aluminum/borate could polymerize isoprene and deliver polyisoprene with low 3,4-selectivity (3,4-content <10%). Later, Visseaux’s group [25] designed substitutive iminopyridine iron catalyst in order to obtain polyisoprene with moderate 3,4-selectivity. However, to the best of our knowledge, only sporadic well-defined iminopyridine iron catalysts had been carried out to afford polyisoprene with 3,4-enhanced structure (3,4-content >50%) with medium activity. Therefore, how to achieve the 3,4-enhanced polyisoprene with high activity by well-defined iminopyridine iron complex remains one of the crucial subjects in the field of academic and industrial processes.

Inspired by the “side-arm” strategy that utilized another coordination site to regulate the activity and selectivity of the catalyst [26,27,28] and based on our group’s ongoing efforts to designing novel well-defined iminopyridine iron complex [29,30,31,32,33,34,35,36], we anticipated that novel iminopyridine iron complex with heterocyclic substituent as side arm might be used to achieve 3,4-enhanced selectivity in isoprene polymerization. Besides that, the response surface method (RSM) as a statistical analytical strategy has been widely employed for optimizing scientific research procedures owing to its being more convenient than the traditional full factorial experiment [37,38,39,40,41,42]. Furthermore, RSM is a helpful way for optimizing and evolving regression equations that check interrelations between different variables, and the uninterrupted mutative surface model could provide the predicted results more closely to the credible experimental data.

In this study, a series of side-arm-assisted tridentate iminopyridine iron complexes were synthesized and characterized, and their catalytic performance toward the 3,4-enhanced polymerization of isoprene with high activity was reported. Response surface methodology (RSM) was employed to elevate and analyze the synergistic influence of polymerization parameters, and the predicted maximal activity via RSM with box-behnken design was matched well with an experimental value under the optimum conditions.

## 2. Materials and Methods

### 2.1. Materials

Use standard Schlenk techniques for all operations with air and/or moisture sensitive compounds. Toluene, dichloromethane (DCM) and n-hexane were purchased from Sinopharm Chemical Reagent, Shanghai, China. Furfurylamine, 2-furanethanamine, 2-thiophenemethylamine and 2-thiopheneethylamine were purchased from Macklin Biochemical Co., Ltd., Shanghai, China. Isoprene and 2-pyridinecarboxaldehyde were purchased from Aladdin Biochemical Technology Co., Ltd., Shanghai, China. Anhydrous ferrous chloride was purchased from Thermo Fisher Scientific Inc., Shanghai, China. Isoprene, toluene, hexane, and dichloromethane were dried over calcium hydride and distilled under argon. All other available reagents were commercial products and were not further purified for the experiment.

### 2.2. Polymerization of Isoprene

In a typical procedure [29], 10.0 μmol iron complex was added to a 25.0 mL Schlenk flask in a glove box. Then, an appropriate amount of toluene was introduced into the flask (V_toluene_:V_Ip_ = 5:2). Subsequently, the flask was moved to the fume hood and then the appropriate isoprene and MAO solution were added into the stirred solution in sequence. The reaction solution was stirred magnetically for a certain time under 25 °C. After that, the reaction was ended by adding 5 mL acidic methanol (V_(methanol)_:V_(HCl)_ = 95:5) to the flask. The polymer was precipitated with methanol and washed several times, and finally dried to a constant weight in vacuum at 60 °C.

### 2.3. Characterization

All the hydrogen nuclear magnetic spectra (^1^H NMR) and carbon nuclear magnetic spectra (^13^C NMR) of ligands were collected on a Bruker Avance III 400 MHz instrument (Bruker, Karlsruhe, Germany) at 298K, using tetramethyl silane (TMS; CIL, Andover, MA, USA) as internal standard. The ^1^H NMR spectra of polyisoprene were tested by NMR instrument with deuterated chloroform as solvent. The number-average molecular weight (*M*_n_) and molecular weight distribution (PDI) of polyisoprene were detected by Agilent GPC instrument (Agilent Technologies Co. Ltd., Shanghai, China) with tetrahydrofuran (HPLC grade) as the eluent at 40 °C (flow rate 1.0 mL per minute). Elemental analysis and mass spectra of iron complex were recorded respectively on a Vario EL III elemental analyzer (Elementar Corporation, Hanau, Germany) at Shanghai Institute of Organic Chemistry (Shanghai, China) and an ACQUITYTM UPLC & Q-TOF MS Premier (Waters, Milford, MA, USA) at Shanghai Jiao Tong University (Shanghai, China).

### 2.4. Synthesis of Iron Complexes

Ligands **L 1–4** were obtained by one step condensation reaction as shown in Figure 2. Briefly, equal molar ratio of 2-pyridinecarboxaldehyde and the corresponding amine compound were introduced into a Schlenk flask with dichloromethane as a solvent and an activated molecular sieve as a water absorbent. The end-up of the reaction process was determined by thin layer chromatography (TLC). The solvent was removed by a rotary evaporator when the reaction was completed, and the target ligand was refined by column chromatography and determined by ^1^H NMR and ^13^C NMR.

The synthesis route of **Fe 1–4** was shown as Figure 2. Complexes (**Fe 1–4**) were obtained by mixing equal molar ratio of FeCl_2_ and **L 1–4** in a 25 mL Schlenk flask with dichloromethane as solvent for 24 h. Upon completion, the solid products were separated from the solution and washed by dried hexane three times. The purity and characterization of the iminopyridine iron (II) derivatives were confirmed by elemental analysis and HRMS.

## 3. Results and Discussion

### 3.1. Analytical Data for Ligands and Iminopyridine Iron Complexes

#### 3.1.1. Characterization of [N, N, O] (**L 1**) and [N, N, O]FeCl_2_ (**Fe 1**)



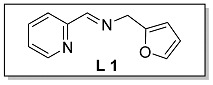



The synthesis route of **L 1** ((E)-N-(furan-2-ylmethyl)-1-(pyridin-2-yl)methanimine) was shown in Figure 2. The specific synthesis steps were as follows. Molecular sieve was added to a 100 mL Schlenk tube and activated at 600 °C for 20 min under vacuum which could remove water and oxygen in reactor. Then 2-pyridinecarboxaldehyde (4.67 mmol, 500.00 mg), furfurylamine (4.67 mmol, 453.34 mg) and dichloromethane (20 mL) were transferred in sequence to the schlenk tube. The reaction lasts for 4 h under room temperature and determined by thin layer chromatography. When the reaction completed, the solu-tion was filtered with diatomite and the dichloromethane was evacuated at room temperature to get the target ligand. yellow oil, 53.1% yield, ^1^H NMR: (400 MHz, Chloroform-d) δ 8.64 (dp, J = 4.0, 1.2 Hz, 1H), 8.42 (d, J = 1.8 Hz, 1H), 8.03 (ddd, J = 8.0, 2.6, 1.2 Hz, 1H), 7.73 (ddd, J = 7.6, 6.2, 2.0 Hz, 1H), 7.39 (tt, J = 1.8, 1H), 7.32 (dddd, J = 8.2, 1 H), 6.35 (h, J = 1.8 Hz, 1H), 6.29 (d, J = 3.0 Hz, 1H), 4.84 (q, J = 1.6 Hz, 2H). ^13^C NMR (100 MHz, CDCl_3_) δ 163.94, 154.48, 151.86, 149.54, 142.48, 136.64, 125.02, 121.52, 110.50, 107.99, 56.95 (see Appendix A).



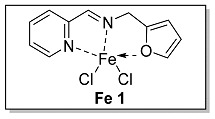



The synthesis route of **Fe 1** was shown in Figure 2. The detailed steps for the synthesis of the catalyst were as follows. In the glove box, the ligand **L 1** (1.07 mmol, 200.00 mg), anhydrous ferrous chloride (1.07 mmol, 136.13 mg) and dichloromethane (30 mL) were transferred to the de-watered and de-oxygenated reaction vials separately and continuously stirred by magnetic stirrer for 24 h. The solution in the mixture was filtered to obtain solid catalyst under argon atmosphere. The catalyst was washed three times by DCM (10 mL × 3) and n-hexane (10 mL × 3) in order, then the solvent was drained and the solid was collected in anhydrous and oxygen-free environment. The catalyst solid powder (**Fe 1**) was collected in a 4 ml glass bottle and stored in a glove box at room temperature. Purple solid powder, 66.2% yield, Elemental Analysis: calcd. for **Fe 1** [C_11_H_10_Cl_2_FeN_2_O + FeCl_2_]: C, 30.05, H, 2.29, N, 6.37; Found: C, 31.05, H, 2.43, N, 6.76. TOF–MS (*m*/*z*) for [ON_2_H_10_C_11_ClFeC_11_H_10_N_2_O]^+^: calcd.: 463.0624. Found: 463.0620.

#### 3.1.2. Characterization of [N, N, O] (**L 2**) and [N, N, O]FeCl_2_ (**Fe 2**)



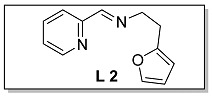



The synthesis route of **L 2** ((E)-N-(2-(furan-2-yl)ethyl)-1-(pyridin-2-yl)methanimine) was shown in Figure 2. The specific synthesis steps were as follows. Molecular sieve was added to a 100 mL Schlenk tube and activated at 600 °C for 20 min under vacuum which could remove water and oxygen in reactor. Then 2-pyridinecarboxaldehyde (4.67 mmol, 500.00 mg), 2-furanethanamine (4.67 mmol, 518.82 mg) and dichloromethane (20 mL) were transferred in sequence to the Schlenk tube. The reaction lasts for 4 h under room temperature and determined by thin layer chromatography. When the reaction completed, the solution was filtered with diatomite and dichloromethane was evacuated at room temperature to get the target ligand. yellow oil, 73.6% yield, ^1^H NMR: (400 MHz, Chloroform-d) δ 8.63 (dt, J = 4.8, 1.2 Hz, 1H), 8.32 (d, J = 1.4 Hz, 1H), 7.96 (dt, J = 8.0, 1.2 Hz, 1H), 7.73 (td, J = 7.8, 1.6 Hz, 1H), 7.34–7.26 (m, 2H), 6.26 (dd, J = 3.2, 1.9 Hz, 1H), 6.05 (d, J = 3.0 Hz, 1H), 3.95 (td, J = 7.2, 1.4 Hz, 2H), 3.07 (t, J = 7.2 Hz, 2H). ^13^C NMR: (100 MHz, Chloroform-d) δ 162.77, 154.56, 153.72, 149.59, 141.29, 136.67, 124.87, 121.46, 110.30, 106.14, 59.68, 29.60.



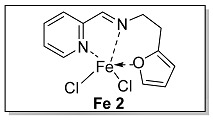



The synthesis route of **Fe 2** was shown in Figure 2. The detailed steps for the synthesis of the catalyst were as follows. In the glove box, the ligand **L 2** (1.00 mmol, 200.00 mg), anhydrous ferrous chloride (0.99 mmol, 127.23 mg) and dichloromethane (30 mL) were transferred to the de-watered and de-oxygenated reaction vials separately and continuously stirred by magnetic stirrer for 24 h. The solution in the mixture was filtered to obtain solid catalyst under argon atmosphere. The catalyst was washed three times by DCM (10 mL × 3) in order, then the solvent was drained and the solid was collected in anhydrous and oxygen-free environment. The catalyst solid powder (**Fe 2**) was collected in a 4 ml glass bottle and stored in a glove box at room temperature. Purple solid powder, 81.0% yield, Elemental Analysis: calcd. for **Fe 2** [C_12_H_12_Cl_2_FeN_2_O]: C, 44.08, H, 3.70, N, 8.57; Found: C, 44.86, H, 3.84, N, 8.79. TOF-MS (*m*/*z*): calcd.: for [ON_2_H_12_C_12_ClFeC_12_H_12_N_2_O]^+^: 491.0937. Found: 491.0941.

#### 3.1.3. Characterization of [N, N, S] (**L 3**) and [N, N, S]FeCl_2_ (**Fe 3**)



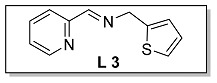



The synthesis route of **L 3** ((E)-1-(pyridin-2-yl)-N-(thiophen-2-ylmethyl) methanimine) was shown in Figure 2. The specific synthesis steps were as follows. Molecular sieve was added to a 100 mL Schlenk tube and activated at 600 °C for 20 min under vacuum which could remove water and oxygen in reactor. Then 2-pyridinecarboxaldehyde (4.67 mmol, 500.00 mg), 2-thiophenemethylamine (4.67 mmol, 528.32 mg) and dichloromethane (20 mL) were transferred in sequence to the schlenk tube. The reaction lasts for 4 h under room temperature and determined by thin layer chromatography. When the reaction completed, the solution was filtered with diatomite and dichloromethane was evacuated at room temperature to get the target ligand. light yellow oil, 43.0% yield, ^1^H NMR: (400 MHz, Chloroform-d) δ 8.65 (ddd, J = 4.8, 1.8, 0.8 Hz, 1H), 8.48 – 8.43 (m, 1H), 8.07 (dt, J = 7.8, 1.0 Hz, 1H), 7.74 (d, J = 1.6 Hz, 1H), 7.32 (ddd, J = 7.4, 4.8, 1.2 Hz, 1H), 7.28–7.22 (m, 1H), 7.02–6.96 (m, 2H), 5.04 (d, J = 1.4 Hz, 2H). ^13^C NMR (100 MHz, CDCl_3_) δ 163.19, 154.50, 149.53, 141.31, 136.66, 127.04, 125.51, 125.06, 125.02, 121.52, 59.04.



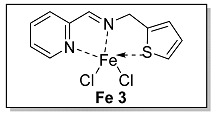



The synthesis route of **Fe 3** was shown in Figure 2. The detailed steps for the synthesis of the catalyst were as follows. In the glove box, the ligand **L 3** (0.99 mmol, 200.00 mg), anhydrous ferrous chloride (0.99 mmol, 125.32 mg) and dichloromethane (30 mL) were transferred to the de-watered and de-oxygenated reaction vials separately and continuously stirred by magnetic stirrer for 24 h. The solution in the mixture was filtered to obtain solid catalyst under argon atmosphere. The catalyst was washed three times by DCM (10 mL × 3) and n-hexane (10 mL × 3) in order, then the solvent was drained and the solid was collected in anhydrous and oxygen-free environment. The catalyst solid powder (**Fe 3**) was collected in a 4 ml glass bottle and stored in a glove box at room temperature. Purple solid powder, 76.7% yield, Elemental Analysis: calcd. for **Fe 3** [C_11_H_10_Cl_2_FeN_2_S]: C, 40.16, H, 3.06, N, 8.51; Found: C, 39.23, H, 3.15, N, 8.22. TOF-MS (*m*/*z*): calcd.: for [SN_2_H_10_C_11_ClFeC_11_H_10_N_2_S]^+^: 495.0167. Found: 495.0166.

#### 3.1.4. Characterization of [N, N, S] (**L 4**) and [N, N, S]FeCl_2_ (**Fe 4**)



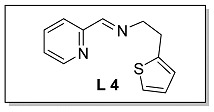



The synthesis route of **L 4** ((E)-1-(pyridin-2-yl)-N-(2-(thiophen-2-yl)ethyl) methanimine) was shown in Figure 2. The specific synthesis steps were as follows. Molecular sieve was added to a 100 mL Schlenk tube and activated at 600 °C for 20 min under vacuum which could remove water and oxygen in reactor. Then 2-pyridinecarboxaldehyde (4.67 mmol, 500.00 mg), 2-thiopheneethylamine (4.67 mmol, 593.79 mg) and dichloromethane (20 mL) were transferred in sequence to the schlenk tube. The reaction lasts for 4 h under room temperature and determined by thin layer chromatography. When the reaction completed, the solution was filtered with diatomite and dichloromethane was evacuated at room temperature to get the target ligand. light yellow oil, 75.2% yield, ^1^H NMR: (400 MHz, Chloroform-d) δ 8.63 (dd, J = 4.8, 1.4 Hz, 1H), 8.32 (d, J = 1.4 Hz, 1H), 8.00 (dt, J = 7.8, 1.2 Hz, 1H), 7.74 (td, J = 7.6, 1.6 Hz, 1H), 7.30 (ddd, J = 7.6, 4.6, 1.4 Hz, 1H), 7.12 (dt, J = 5.0, 1.2 Hz, 1H), 6.91 (dd, J = 5.0, 3.4 Hz, 1H), 6.85 (dd, J = 3.4, 1.2 Hz, 1H), 3.95 (td, J = 7.0, 1.4 Hz, 2H), 3.27 (t, J = 7.0 Hz, 2H). ^13^C NMR (100 MHz, CDCl_3_) δ 162.83, 154.57, 149.56, 142.21, 136.64, 126.84, 125.24, 124.84, 123.76, 121.44, 62.65, 31.36.



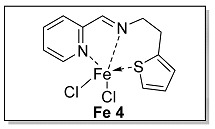



The synthesis route of **Fe 4** was shown in Figure 2. The detailed steps for the synthesis of the catalyst were as follows. In the glove box, the ligand **L 4** (0.93 mmol, 200.00 mg), anhydrous ferrous chloride (0.93 mmol, 117.74 mg) and dichloromethane (30 mL) were transferred to the de-watered and de-oxygenated reaction vials separately and continuously stirred by magnetic stirrer for 24 h. The solution in the mixture was filtered to obtain solid catalyst under argon atmosphere. The catalyst was washed three times by DCM (10 mL × 3) and n-hexane (10 mL × 3) in order, then the solvent was drained and the solid was collected in anhydrous and oxygen-free environment. The catalyst solid powder (**Fe 4**) was collected in a 4 ml glass bottle and stored in a glove box at room temperature. Purple solid powder, 63.1% yield, Elemental Analysis: calcd. for **Fe 4** [16C_11_H_10_Cl_2_FeN_2_S+CH_2_Cl_2_]: C, 41.59, H, 3.51, N, 8.04; Found: C, 42.56, H, 3.78, N, 8.13. TOF-MS (*m*/*z*): calcd.: for [SN_2_H_12_C_12_ClFeC_12_H_12_N_2_S]^+^: 523.0480. Found: 523.0486.

### 3.2. Optimization of Single Factors

All these iron complexes were utilized as pre-catalysts to evaluate the performance of isoprene polymerization. Firstly, the influence of different heteroatoms and the number of linkers in the iron complex were optimized, and the results are shown in Table 1. The special side-arm [N, N, X] iminopyridine iron complex exhibited outstanding activity for isoprene polymerization and produced 3,4-enhanced polyisoprene. By comparing the activity of **Fe 1** (8.0 × 10^5^ g·mol(Fe)^−1^·h^−1^) and **Fe 3** (7.9 × 10^5^ g·mol(Fe)^−1^·h^−1^), it could be concluded that the effects of different heteroatom on the activity of tridentate catalysts were consistent with the coordination ability of heteroatom substituent (furan ≈ thiophene). In addition, the 3,4-selectivity (**Fe 1**: 60% vs. **Fe 3**: 62%) and the *M*_n_ (**Fe 1**: 38.0 × 10^4 v^g/mol vs. **Fe 3**: 42.7 × 10^4^ g/mol) of polymer catalyzed by **Fe 1** and **Fe 3** were also almost the same. On the other hand, the number of carbon atoms linked between imine and its substituent was also investigated. Upon activated by 500 equivalent MAO, the [N, N, X] iminopyridine iron complex with two carbon linkers (**Fe 2** and **Fe 4**) exhibited slightly higher catalytic activity, lower 3,4-selectivity and *M*_n_ of polyisoprene than that with one carbon linker (**Fe 1** and **Fe 3**). It might be explained by the fact that the increase in methylene leads to a decrease in the steric hindrance of the active center, and isoprene is easier to coordinate. Among these complexes, **Fe 2** showed higher catalytic activity than others; therefore, **Fe 2** was selected as the main catalyst to conduct the following single-factor optimization research.

In order to comprehensively investigate the main factors affecting the activity under room temperature, the reaction time, Ip/Fe ratio, and Al/Fe ratio were selected for single-factor optimization. Initially, various reaction times were carried out; the results are shown in Table 2. The reaction time was varied from 10 to 1 min (Table 2, entries 1–3) when the Al/Fe ratio was fixed at 500. The full conversion could be achieved at 1 min, which illustrated that **Fe 2** possesses excellent catalytic activity (8.2 × 10^6^ g·mol(Fe)^−1^·h^−1^). Meanwhile, the 3,4-selectivity of polyisoprene and the *M*_n_ of polyisoprene were changed inconspicuously with reaction time. To purchase higher activity, the IP/Fe ratio was investigated as well. It was observed that when the IP/Fe ratio increased from 2000 to 6000, the activity improved from 8.2 × 10^6^ g·mol(Fe)^−1^·h^−1^ to 2.3 × 10^7^ g·mol(Fe)^−1^·h^−1^ without apparent change in the 3,4-selectivity and the molecular weight. Further increasing the IP/Fe ratio to 8000, the activity was slightly reduced from 2.3 × 10^7^ g·mol(Fe)^−1^·h^−1^ to 2.2 × 10^7^ g·mol(Fe)^−1^·h^−1^. Finally, the Al/Fe ratio was examined (Table 2, entries 5 and 7–9). It was known that the activity (2.5 × 10^7^ g·mol(Fe)^−1^·h^−1^ to 2.3 × 10^7^ g·mol(Fe)^−1^·h^−1^) and the 3,4-selectivity (57% to 56%) were nearly unchanged when the Al/Fe ratio varied from 1000 to 500. However, further decreasing Al/Fe ratio to 250, the activity was slightly decreased from the 2.3 × 10^7^ g·mol(Fe)^−1^·h^−1^ to 2.1 × 10^7^ g·mol(Fe)^−1^·h^−1^ with 85% conversion. The reason might be that the impurities in the solvent and monomer expended some MAO, which resulted in the remaining MAO being insufficient to activate all pre-catalysts. Similarly, 3,4-selectivity had unobvious alterations when Al/Fe ratio changed the result from excessive MAO mainly acting as an alkylation and dealkylation reagent.

### 3.3. Response Surface Methodology (RSM) Design and Experiments

Describing and possibly predicting the results of nonlinear dependence on multivariate conditions can be achieved through a design of experiments, which is a widely known procedure for testing hypotheses [42]. Box–Behnken Design (BBD) was conducted with three independent variables (*X_1_*, Al/Fe; *X_2_*, IP/Fe; *X_3_*, reaction time) as demonstrated in Table 3. The levels of independent variables were selected on the basis of the preliminary research with these catalytic systems. The polymerization activity (*Y*) was chosen as the only response parameter of the designed experiment. The definition of *Y* was described as Equation (1), and the predicted activity was confirmed by multiple regressions to suit the quadratic polynomial model as Equation (2), wherein in Equation (1), *M_p_* is the weight of the polymer, *t* is the reaction time, and *n_Fe_* is the molar of the iron complex.(1)$$Y=\frac{M_{p}}{t\cdot n_{Fe}}$$


While in Equation (2), *Y* is the predicted activity, and *β_0_* is an intercept, *β_i_*, *β_ij_*, and *β_ii_* are regression coefficients for linear, quadratic, and interactive terms, respectively. *X_i_* and *X_j_* are the coded independent variables, respectively.
(2)$$Y=\beta_{0}+\sum_{i=1}^{3}\beta_{i}X_{i}+\sum_{i<j}^{2}\sum_{j}^{3}\beta_{ij}X_{i}X_{j}+\sum_{i=1}^{3}\beta_{ii}X_{i}^{2}$$

Based on the foregoing single-factor results, a 17-run BBD isoprene polymerization experiments catalyzed by tridentate iminopyridine iron complex were established to intensively optimize *X_1_* (Al/Fe), *X_2_* (Ip/Fe), and *X_3_* (reaction time). The experimental conditions and corresponding catalytic activities of polymerizations are summarized in Table 4. The coefficient of determination (R^2^) was utilized to evaluate the reliability of the created model, while the data of multiple regression and mean square of residual error were used to assess the statistical results of the model by analysis of variance (ANOVA). As shown in Table 5, the F-value is 18.29, indicating that the optimized model is significant. Meanwhile, the value of R^2^ is 0.9592, and the adjusted R^2^ is 0.9068, which is also confirmed to be statistically significant. Additionally, the value of the signal-to-noise ratio was 13.2168, which was remarkably greater than 4, indicating the established model with high accuracy. The *p*-value is taken as an effective measure to inspect the significance of coefficients between each independent variable as well. It could be discovered that the coefficients of *X*_3_ were extremely significant (*p* < 0.001), and the coefficients of *X*_1_, *X*_1_
*X*_2_, and *X*_3_^2^ were found to be significant (*p* < 0.05) as well.

The contour and 3D response surface diagrams were obtained by Design Expert Software (shown in Figure 1 and Figure 2) based on Box–Behnken design experiment results. In these two drawings, the catalytic activity of polyisoprene was obtained with continuous variables, while the other parameters were fixed at the optimum value. The optimal reaction conditions to achieve the maximum activity of isoprene polymerization were as follows: the ratio of Fe/Al/IP was 1/683/7095, the reaction time was 0.52 min (31.2 s), and the predicted value of activity was 3.9863 × 10^7^ g·mol(Fe)^−1^·h^−1^. Under the same conditions, the experimental activity was 4.0889 × 10^7^ ± 2.6% g·mol(Fe)^−1^·h^−1^ (number of repeated experiments (n) = 3; see Table 6), which conformed with the prediction of activity. It means that the model is competent for the aggregation process.

## 4. Conclusions

In this study, tridentate iminopyridine iron complexes with substituted ligands of furan-[N, N, O] and thiophene-[N, N, S] as side-arm have been synthesized. The results of HRMS and elemental analysis showed that the complexes (**Fe 1–4**) were successfully prepared. The iron complex **Fe 1–4** was highly active for isoprene polymerization and give 3,4-enriched polyisoprene. The highest 3,4-selectivity (up to 62%) was obtained by **Fe 3** which thiophene group used as side arm. What’s more, RSM was applied in the optimization activity of polymerization by **Fe 2**, and an extremely high activity (4.0889 × 10^7^ g·mol(Fe)^−1^·h^−1^) was achieved. It was observed that the predicted maximal activity by response surface methodology was consistent with the experimental activity. Therefore, this work represents not only the first report about tridentate iminopyridine iron complexes with side-arm catalyzed 3,4-enhanced isoprene polymerizations with high activity but also about the first application of RSM in the field of well-defined iminopyridine iron complexes catalyzed conjugated dienes polymerizations.

## Data Availability

The data presented in this study are available upon request from the corresponding author.

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
