# Peer review of "3,4-Enhanced Polymerization of Isoprene Catalyzed by Side-Arm Tridentate Iminopyridine Iron Complex with High Activity: Optimization via Response Surface Methodology"

_polymers, 2023, doi:10.3390/polym15051231_

Round 1

Reviewer 1 Report

This paper by Wang et al. presents iron complexes for diene polymerisation. The idea of using iron complexes for polymerisation is based on literature data. It is worth mentioning that Ritter et al. demonstrated the possibility of isoprene polymerization with iminopyridine iron complex activated by alkyl aluminium/borate. In this work a slight influence of heterocyclic substituents in the structure of iminopyridine was supposed to replace the activation of the iron complex according to the authors. One drawback of this work is the lack of 13C spectra for the organic ligands, which would unambiguously confirm the structure of the putative molecules. In addition there is no structural evidence of the resulting complexes, no XRD for at least one of them. This remark refers to the illusory involvement of the sulphur and oxygen atoms of the heterocycles in coordination with the iron ion. The part of the paper dealing with diene polymerization is sufficiently detailed and does not raise significant doubts. Thus, I recommend the authors to add the necessary data concerning organic ligands and complexes based on them.

Reviewer 2 Report

The manuscript reports on the polymerization of isoprene catalyzed by iron complexes of iminopyridines with different side arms. 

This is an interesting work but it needs some correction in the format and presentation. On the other hand there are many mistyping errors. 

The format of citations in the text do not suit to the journal's style (they must not put into superscript), and the format of reference list is not completely uniform.

In page 2, line 55-60 the sentence is too long and difficult to understand. Furthermore, the following parts are also difficult to understand:

"were shown similar phenomenon" (page 5, line 163)

"3,4-enriched structure by activated with MAO" (page 9, line 256)

Write in more details that the activator MAO how can have influence on the reactions.

In Scheme 2 the first step is a condensation reaction and complete with -H2O under the reaction arrow.

In Equation 1 the denotation mFe is incorrect if it is related to the molar quantity, use nFe instead of it. 

Nothing can be found in the text about what the numbers 0, 1, -1 in Table 4 mean exactly. 

Round 2

Reviewer 1 Report

I am grateful to the authors for their work on the corrected version of the manuscript. Only one point remains unresolved. Without structural proof and without any examples from the literature, one cannot claim that the heterocyclic residues in the ligands are involved in coordination with the metal ions. This is not a decisive point at this stage; however, the authors may pay much attention to this point in the future.